# Unconventional mechanical and thermal behaviours of MOF CALF-20

Dong Fan [1,2], Supriyo Naskar [1] & Guillaume Maurin [1] ✉

CALF-20 was recently identified as a benchmark sorbent for $CO_2$ capture at the industrial scale, however comprehensive atomistic insight into its mechanical/thermal properties under working conditions is still lacking. In this study, we developed a general-purpose machine-learned potential (MLP) for the CALF-20 MOF framework that predicts the thermodynamic and mechanical properties of the structure at finite temperatures within first-principles accuracy. Interestingly, CALF-20 was demonstrated to exhibit both negative area compression and negative thermal expansion. Most strikingly, upon application of the tensile strain along the [001] direction, CALF-20 was shown to display a distinct two-step elastic deformation behaviour, unlike typical MOFs that undergo plastic deformation after elasticity. Furthermore, this MOF was shown to exhibit a fracture strain of up to 27% along the [001] direction at room temperature comparable to that of MOF glasses. These abnormal thermal and mechanical properties make CALF-20 as attractive material for flexible and stretchable electronics and sensors.

The flexible Metal-Organic Frameworks (MOFs) also called soft porous crystals or dynamic MOFs have emerged as a smart subclass of MOFs owing to their versatile physical properties that challenge conventional conceptions of framework rigidity/fragility in material science[1–4]. In particular the intriguing mechanical and thermal properties of this family of materials serve as an important cornerstone in expanding their potential application domains[3,5–7]. Notably, some of the flexible MOFs, possess counterintuitive mechanical properties, including negative linear compressibility (NLC)[8], negative area compressibility (NAC), stretch densification[9], and push/pull-twisted among others[10]. These abnormal pressure-responsive behaviour defies the well-known "compression-contraction" effect and paves the way for many applications, e.g. optoelectronic devices[11], pressure sensors, and intelligent body armour[9,12]. In addition, some of these materials also exhibit negative thermal expansion (NTE) behaviour, which refers to the phenomenon where a material contracts rather than expands upon heating, of potential importance for further development in the field of thermal expansion compensator.

Recently, a $Zn_2(1,2,4$-triazolate$)_2$(oxalate) framework named CALF-20, was proposed as a benchmark sorbent for $CO_2$ capture combining excellent $CO_2/N_2$ separation performance in flue gas conditions, high hydrolytic stability and easy scalable that makes it transferable at the industrial scale[13]. The structure of this MOF consists of 2D layers of 1,2,4-triazolate-bridged zinc ions pillared by oxalate linkers, forming a three-dimensional (3D) framework[13]. The pillars of the oxalate linker acting as hinges, are connected by zinc triazolate grids parallel to the **ac** plane, offering potentially high flexibility to the 3D framework that might pave the way towards intriguing physical properties (Fig. 1a). Other CALF-20 derivatives, as well as their composites, have been equally explored[14–19]. However, to date, the mechanical/thermal behaviours of CALF-20 have been overlooked both experimentally and theoretically. Although several studies have been reported on the competitive adsorption of $CO_2$ and $H_2O$ in CALF-20 and derivatives[19–21] the lack of understanding on the pressure/thermal-induced structure evolution of CALF-20 is critical in the context of using this MOF under diverse working conditions. The determination of MOF mechanical/thermal properties generally requires the deployment of sophisticated experimental techniques as well as the consideration of high-quality samples making the whole testing process far from trivial[22,23]. Meanwhile, in-silico simulation approaches offer promising alternatives to not only accurately predict but also gain atomistic insight and even rationalise the MOF structure

[1]ICGM, Univ. Montpellier, CNRS, ENSCM, Montpellier 34095, France. [2]School of Materials Science and Engineering, Chongqing Jiaotong University, Chongqing 400074, PR China. ✉e-mail: guillaume.maurin1@umontpellier.fr

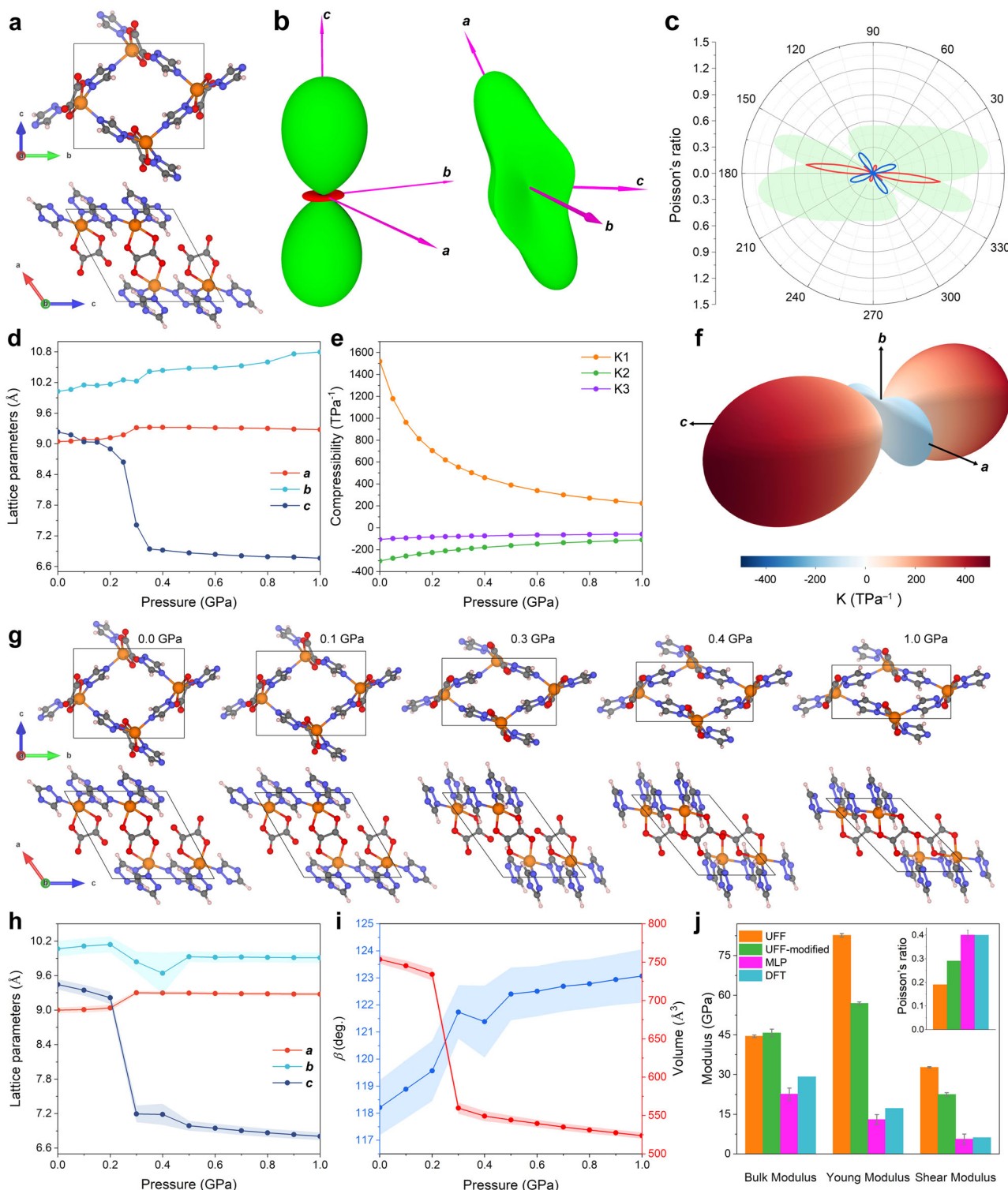

**Fig. 1 | Anisotropic mechanical properties of CALF-20 predicted by DFT and finite temperature NPT MLP-MD simulations. a** DFT-optimised structure of CALF-20 viewed along different perspectives. Colour code: Zn, orange; N, blue; O, red; C, grey; H, white. **b** Three dimensional representations of the linear compressibility (left) and Young's modulus (right). Positive and negative values for linear compressibility are indicated as green and red respectively. **c** The elastic constant derived from the spatial dependence of the Poisson's ratio in **ac** plane. Positive and negative contributions are indicated in blue and red lines, respectively. The green shadow represents the maximum positive value for Poisson's ratio. **d** Unit-cell parameters changes and corresponding **e** compressibility changes during hydrostatic compression of CALF-20. **f** 3D mapping of the linear compressibility (TPa$^{-1}$) derived using the structural parameters under different mechanical pressures as determined by an empirical potential fitting approach[31]. **g** Evolution of the CALF-20 unit-cell computed by DFT at different pressures. **h, i** Structural parameters and unit-cell volume changes with applied mechanical pressure computed by MLP-MD simulations runs in the NPT ensemble at room temperature (298.15 K). The error bars represent the statistical deviation obtained from 10,000 MD trajectories collected over the last 0.5 ns of the MD runs. **j** Overall elastic properties simulated by MD calculations using different force fields, including MLP as well as UFF and UFF-modified referring to an adjusted UFF parameterisation reported in our previous work[18], compared to DFT calculations at 0 K.

behaviours upon diverse environments[24–26]. In-depth computational exploration of the physical properties of the emblematic CALF-20 MOF calls for the development and application of cost-effective and accurate molecular simulation tools. The expected outcomes might reveal interesting phenomena paving the way towards innovative prospective for this MOF.

In this context, the mechanical and thermal behaviours of CALF-20 was scrutinised by combining extensive Density Functional Theory (DFT) calculations and high-precision MLP-MD simulations. CALF-20 was demonstrated to exhibit anomalous structural response to both temperature and pressure stimuli, resulting in the coexistence of NAC and NTE phenomena. This makes this material highly attractive for mechanical/thermal sensing, besides its actual application in the field of $CO_2$ capture. Decisively our simulations revealed that CALF-20 exhibits an abnormal strain-softening at both 0 K and room temperature under uniaxial tensile strain that eventually leads to the formation of a metastable structure. From a methodology perspective, the MLP-MD approach was demonstrated to be a powerful tool for anticipating unconventional structural features of CALF-20 upon stimuli at finite temperature.

## Results and discussion
### Negative area compressibility behaviour
DFT calculations were first performed at 0 K to determine the elastic properties of CALF-20 via a comprehensive tensorial analysis based on a finite difference approach[27], as implemented in the VASP code[28]. These calculations evidenced that CALF-20 exhibits a large anisotropy in its linear compressibility, with a positive contribution along the *c* axis (170 TPa$^{-1}$) (see Fig. 1b), while both *a* and *b* axis show a negative contribution (maximum of −24 TPa$^{-1}$ and −32 TPa$^{-1}$ respectively, see Supplementary Fig. 1). This unusual mechanical response of the MOF framework corresponds to a rather rare NAC behaviour. This implies that as the MOF is compressed, the *ab*-plane expands spontaneously to maintain the minimum energy principle of the system, while the *c*-axis shrinks. These calculations revealed that the predicted Young's modulus (Fig. 1b) and Poisson's ratio (Fig. 1c) are also highly anisotropic, with both maxima observed along the *a*-axis (Supplementary Fig. 2) corresponding to the direction of the pillared oxalate ligands, allowing a stronger resilience of the MOF framework, whereas the *bc* plane corresponds to the more flexible rhombic-shaped zinc triazolate grid. It is worth noting that the [100] and [010] directions are the principal axes of the rhombic shaped zinc triazolate grid, therefore, when *b*-axis is elongated, the zinc triazolate layer is deformed by reducing the length of all the connectors along the *c*-axis to minimise the energy of the overall system. Therefore, the unique flexible behaviour of CALF-20 originates from its topology. Each oxalate ligand utilises its tetradentate oxygen atoms to vertically stretch two independent zinc triazolate ligand forming a crossed 3D configuration with zinc triazolate grids and oxalate pillars.

Besides the stiffness tensor-based analysis, the impact of applying a mechanical pressure on the structure was equally examined. Compressibility of material is typically denoted as the relative rate of dimension changes with pressure at a fixed temperature, $\kappa = -\frac{1}{i}\frac{d_i}{d_p}$, where *i* can be assigned as volume, area, or linear compressibility, respectively[9]. The compressibility coefficients of the principal axes can be determined within the pressure range[29]. Figs. 1d-1e show that the most striking giant NAC phenomenon along *a* and *b* axis continues monotonically up to 0.3 GPa. The NAC behaviour is much more marked along the *a*-axis with −188.77 TPa$^{-1}$ strain tensor eigenvector compared to that along the *b*-axis (−75.52 TPa$^{-1}$). The highest positive linear compressibility (PLC) is observed along the direction perpendicular to the *ab* plane with a corresponding eigenvector of 501.53 TPa$^{-1}$, which leads to a substantial compression of the oxalate pillars under pressure (cf. Figure 1f, Supplementary Table 1). Remarkably, in contrast to other conventional uniaxial NLC materials, CALF-20

exhibits a biaxial (or in-plane) NLC behaviour, and its compressibility is higher than that of the most extreme existing NLC inorganic materials including typical ferroelastics (−0.137 ~ −4.3 TPa$^{-1}$)[30], $Ag_3[Co(CN)_6]$-I ($K_c = -76$ TPa$^{-1}$)[31] and $[Ag(en)]NO_3$-I ($K_c = -28.4$ TPa$^{-1}$)[32], as well as the emblematic flexible MIL-53 MOF material ($K_b = -27$ TPa$^{-1}$)[33]. The CALF-20 structures simulated upon different applied mechanical pressure are illustrated in Fig. 1g. It is evident that as the pressure increases, the *c*-axis shrinks significantly, eventually leading to a decrease in unit cell volume derived from the larger PLC along the axis. Notably, the applied pressure up to 1 GPa does not induce any structure collapse. Therefore, besides its previously-demonstrated promise as $CO_2$ sorbent at the industrial level, CALF-20 owing to its unconventional mechanical properties, is predicted to be attractive in pressure-sensitive devices, shock-absorbing materials, artificial muscles, and shape-memory applications among others. Supplementary Figs. 3-4 that reports the pressure-induced evolution of the CALF-20 lattice parameters further demonstrates that above 1 GPa, a metastable high-pressure phase of CALF-20 exists. Supplementary Fig. 3 shows that such a phase is non-porous and associated with a very high density (3.58 g cm$^{-3}$). This phase is also demonstrated to remain stable even after releasing the pressure stimulus (Supplementary Fig. 4).

We then developed an MLP-MD strategy (Supplementary Note 1 and Supplementary Figs. 5–14 and Supplementary Movies 1-2 for details) to explore the pressure-driven structural flexibility of CALF-20 at room temperature, which is almost unfeasible within the DFT formalism. All MLP-MD simulations were run for 1 nanosecond (ns) with 0.5 femtosecond (fs) time steps with the consideration of a $2 \times 2 \times 2$ supercell for CALF-20. Figure 1h, i report the MLP-MD derived evolution of the CALF-20 lattice parameters at room temperature under different pressures. These simulations evidenced that both *a*- and *b*-axis dimensions decrease with increasing pressure below 0.30 GPa in line with the NAC phenomenon revealed by DFT calculations at 0 K. An apparent phase transition is equally observed around 0.30-0.4 GPa (Figs. 1d and 1h). The fluctuations of the *b*- and *c*-axis dimensions are found to be larger than that of the *a*-axis one within the phase transition region as shown in Fig. 1h. This is mainly due to the intrinsic anisotropic flexibility of the CALF-20: its *bc*-plane corresponds to the more flexible zinc triazolate lattice, while the a-axis is aligned with the more rigid pillared oxalate ligands. Moreover, when pressure exceeds 0.50 GPa, *a*- and *b*-axis dimensions remain almost unchanged upon pressure increase, while the c-axis dimension decreases linearly (Fig. 1h). These MLP-MD simulations suggest that CALF-20 is expected to display at room temperature a tiny NAC behaviour above 0.50 GPa, where the structure evolves as a rigid backbone. The reliability of MLP vs generic UFF force field to describe the mechanical properties of CALF-20 was further assessed using DFT results as benchmark data. Figure 1i and Supplementary Table 2 show that UFF force field leads to a substantial overestimation of both bulk modulus and Young's modulus while it significantly underestimates the shear modulus by a factor of 3 and 4, respectively, compared to the corresponding DFT value. Remarkably, the use of MLP enables to reproduce well the DFT values, especially, for the Young's modulus and shear modulus. This comparison emphasises the reliability of the MLP-MD approach to accurately probe the flexible behaviour of CALF-20 at finite temperature at a computational cost far below that of DFT calculations (Supplementary Note 2).

### Negative thermal expansion behaviour
The temperature-dependent thermal properties of CALF-20 were first assessed using the quasi-harmonic approximation (QHA) approach[34], where the influence of temperature is considered through the volume response of the vibrational frequency via the use of phonon anharmonicity[35]. Note that conventional QHA approach based on DFT calculations requires the determination of the phonon spectrum under several independent volume deformation of the structure, which

requires huge computing resources. The implementation of MLP in the QHA scheme enables a much faster assessment of the corresponding data reported in Fig. 2a (see Supplementary Figs. 15, 16 for details). CALF-20 is thus predicted to exhibit a NTE behaviour at low temperature, the lowest NTE value of $-10.56 \times 10^{-6}$ K$^{-1}$ at 40 K being within the same range than that reported for typical NTE MOFs[36–38]. Notably, a negligible unit cell volume change versus temperature is obtained, indicating that the lattice shrinks in different orientations as the temperature increases to ensure an energy equilibrium, confirming a relatively high flexibility of the CALF-20 framework. Since QHA does not explicitly include anharmonicity[39], its reliability is almost exclusively limited to the low-temperature domain. One possibility to extend the applicability of QHA to higher temperature is to carry out ab-initio molecular dynamics (AIMD) simulations, however, this is time-consuming. Herein, we privileged the implementation of our MLP validated above on the mechanical properties of CALF-20 to run MLP-MD simulations in the NPT ensemble at 1 bar and room temperature (298.15 K) with a total simulation time of 1 ns with the use of 0.5 fs time steps to determine the equilibrium lattice parameters of CALF-20, as shown in Fig. 2b, c. We evidenced that the **b**-axis dimension decreases linearly with increasing temperature while the overall unit-cell volume remains almost constant, showing a unique NTE feature[40], in excellent agreement with the prediction based on the QHA methodology. Figure 2d delivers a 3D-representation of the contribution to thermal expansion along each axis (also see Supplementary Table 3). The blue region refers to the NTE behaviour domain with negative linear thermal expansion ranging from −4.22 MK$^{-1}$ and −60.81 MK$^{-1}$. While this coefficient is comparable to that exhibited by MOF DUT-60 (− 65.0 MK$^{-1}$)[41] it is twice higher than the values reported for other typical NTE MOFs, e.g. Ag(mim) (− 24.5 MK$^{-1}$)[42], and DUT-49 (− 32.778 MK$^{-1}$)[43]. This suggests CALF-20 is a promising candidate for applications in the fields of ultrasensitive thermal sensing.

## Abnormal tension strain response

Typically, a material with multiple negative phenomena, e.g. NLC combined with NTE exhibits other peculiar strain-stress behaviours[44,45]. We therefore examined the deformation behaviour of CALF-20 upon the application of ideal tensile strain. The strain-stress curve was first simulated at 0 K by DFT calculations (Fig. 3a). The three directions show anisotropic changes, among which the [100] and [010] directions show rapid strain-stiffening after a short elastic deformation[46,47]. As a result, the [100] and [010] growth directions are predicted to be relatively brittle with failure strains lower than 11% and 22% respectively. However, two independent elastic deformation regions were found with the tensile strain along [001] direction, namely before and after strain-softening to MOF fracture respectively. The first peak at 18.43% strain is associated with a structural transition, i.e. strain-softening point, while the second one at 40.25% corresponds to MOF fracture. The rapid change in the stress-strain curve suggests a phase transition. Therefore, the structure near the strain-softening point was considered to calculate the phonon spectrum. Figure 3b, c and Supplementary Figs. 17–19, reveal that there is no any imaginary frequency in the entire Brillouin zone and this confirms the dynamic stability of CALF-20 during the deformation process. In the meantime, we demonstrated that this MOF maintains its structure integrity at very high strain level up to 36.13% (Supplementary Fig. 20). This behaviour significantly differs with that exhibited by most of the MOF crystals like Ni-TCPP[48] and ZIF-8[49] that all undergo a fracture above 30% strain.

MLP-MD calculations were further deployed to explore the temperature effect on the strain-stress behaviour of CALF-20. Figure 3d, e show the MOF structure response upon the application of the tensile strain along the three directions at 200 K and 298.15 K, respectively. These MLP-MD simulations reproduce very well the trend obtained by DFT calculations at 0 K while UFF dramatically fails (see Supplementary Figs. 21–23 for details). In particular, MLP-MD

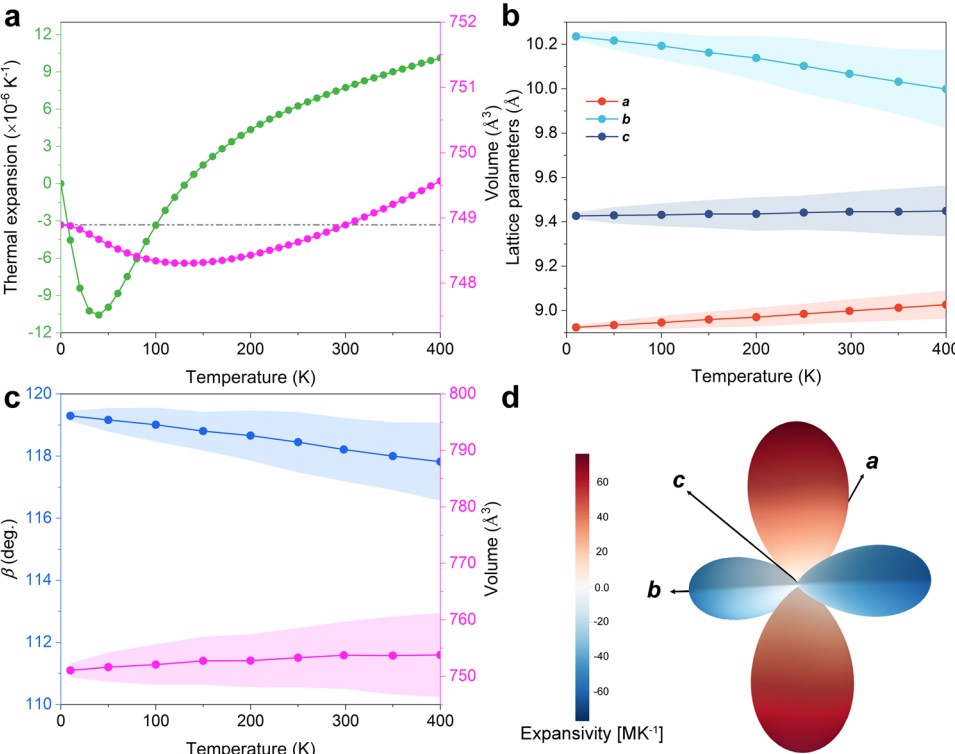

**Fig. 2 | Anisotropic Negative Thermal Expansion of CALF-20 revealed by finite temperature NPT MLP-MD simulations. a** Thermal expansion coefficient and volume change calculated at different temperatures using the quasi-harmonic approximation. The grey dashed line represents the volume at 0 K. **b-c** Equilibrium lattice parameters calculated at various temperatures based on MLP-MD simulations. The error bars indicate the statistical deviation from the collected trajectories. **d** Illustration of the 3D expansivity (MK$^{-1}$) derived from the empirical potential fitting method using MLP-MD outputs.

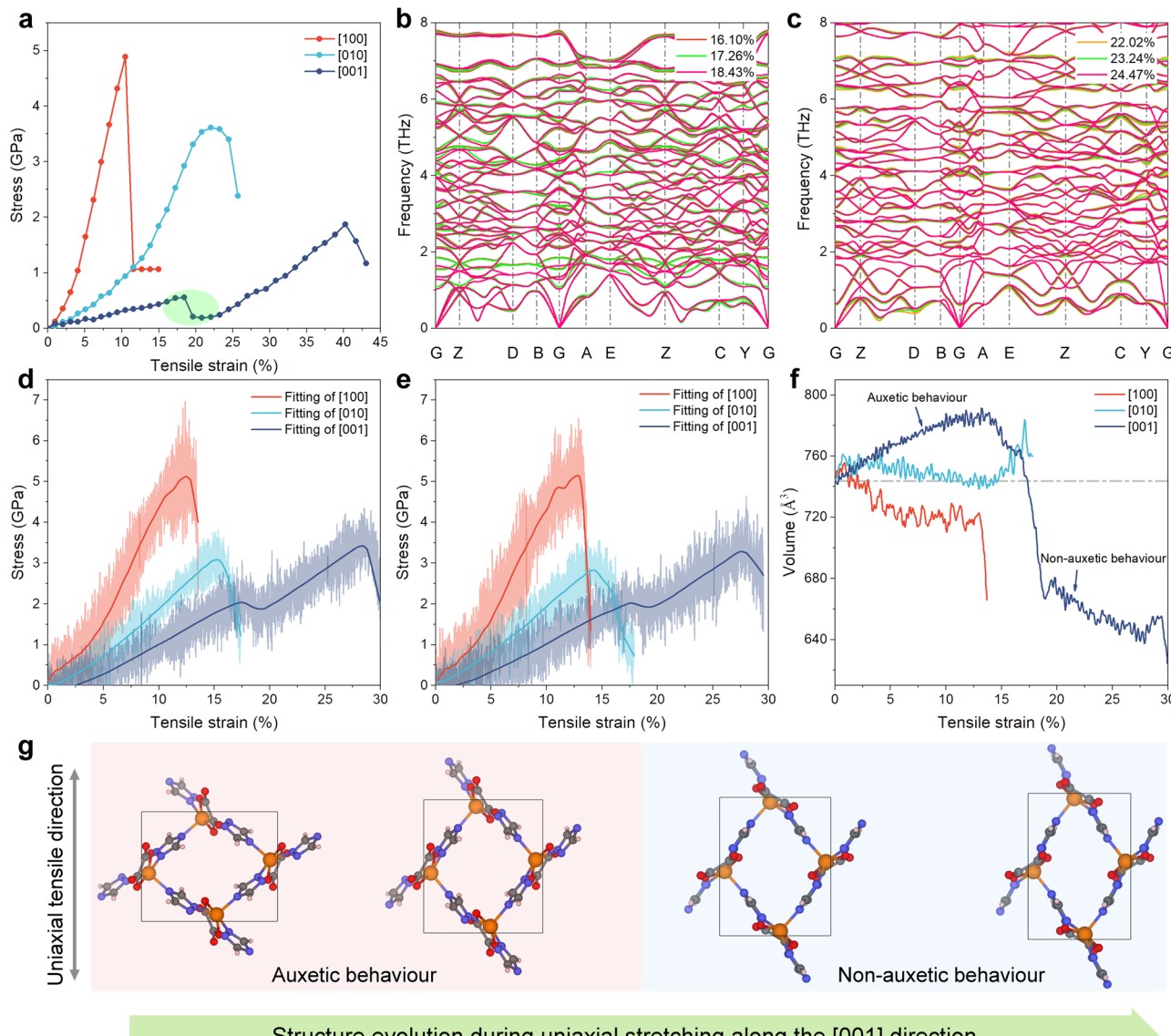

**Fig. 3 | Unconventional strain-stress behaviour determined by DFT and finite temperature NPT MLP-MD simulations. a** Tensile strain-stress response along the three directions from DFT calculations at 0 K. The green circle indicates the strain-softening region. **b**, **c** Phonon dispersion spectrum under tensile strain (around the strain-softening range) along the [001] direction calculated based on MLP. Only below 8 THz curves are displayed for clarity. Stress-strain curves at (**d**) 200 K and (**e**) 298.15 K derived from MLP-MD simulations. **f** MLP-MD derived volume change of CALF-20 at 298.15 K upon application of tensile strain along the three directions. The dotted line represents the unit-cell volume of the pristine CALF-20 structure. The two independent auxetic and non-auxetic phenomena upon stretching along the [001] direction are also highlighted. **g** Schematic illustration of the CALF-20 structural evolution upon tensile strain applied along the [001] direction. Colour code: Zn, orange; N, blue; O, red; C, grey; H, white. The auxetic and non-auxetic behaviours correspond to the two regimes observed in Fig. 3f.

predicts specific stress very close to the DFT-simulated value. These calculations further revealed that the failure strain of CALF-20 remains as high as 27%, even at room temperature. To our knowledge, no MOF crystal has ever demonstrated such a wide range of elastic deformation behaviour at finite temperature, and the failure is comparable to that of MOF glasses (20% ~ 35% fracture strain recently reported for a$_g$ZIF-62)[50,51]. On the other hand, we found that the temperature effect on the fracture strain of CALF-20 is very small, *e.g.*, the fracture strain along the [100] direction is almost unchanged when the temperature increases from 200 K to room temperature, and the [010] and [001] direction dimensions are reduced by only 1%. This trend is likely to be due to the coexistence of the inherent NTE and NAC behaviours of CALF-20 that makes its framework resilient to the deformation process by modulating its own flexibility during the deformation process.

Figure 3f reports the unit cell volume changes upon tensile strain applied in the three directions at room temperature (see the corresponding profiles at 200 K and 0 K in the Supplementary Figs. 24, 25, respectively). It is clear that when the tensile strain is applied along the [001] direction (before the strain-softening point), CALF-20 shows a very different unit cell volume expansion behaviour than in the other directions. This is attributed primarily to the significant Negative Poisson Ratio (NPR) obtained during this process, the lowest value being of −0.35, as shown in Fig. 1c and Supplementary Fig. 12). Consequently, the constraint applies along the [001] direction and the structure expands along the *a*- and *b*-axis and then display auxetic behaviour, as shown in Fig. 3g and Supplementary Fig. 26. When the tensile strain of the MOF along the [001] direction exceeds the strain-softening point, the phase transition occurs and the NPR feature disappears, showing non-axial behaviour.

## Reversible strain-induced phase transition mechanism

To further explore the structure deformation mechanism of CALF-20 under different strains, we defined three deformation angles by considering the Zn metal as the node of the deformation framework and two different ligands as the rigid rods, as shown in Fig. 4a. Among them, $\phi$ and $\theta$ are defined as the angles of the lozenge lattice formed by triazolate grids, and $\omega$ represents the angle between the triazolate grids and oxalate ligands. As shown in Supplementary Fig. 27 and Figs. 4b, c, all the angular changes of CALF-20 are linear before the critical strain regardless of the strained direction. This implies that the MOF is sufficiently robust to resist to mechanical deformation. Notably, the deformation proceeds similarly before and after the strain softening-point when the tensile strain is applied along the [001] direction. This suggests that the CALF-20 structure presents a high flexibility in conjunction with a good mechanical strength. In particular, the linear relationship between the angular changes and the applied tensile strain is still preserved above 30% strain as shown in Fig. 4c and Supplementary Movie 3. This MOF offers alternative

solution to Mxene and graphene[52,53] for ultra-sensitive strain-sensing applications since its predicted elongation range is lower than 5%. To further elucidate the phase transition mechanism driven by the uniaxial tensile strain along the [001] direction, we conducted orbital Hamilton population analysis[54], which allows quantitative analysis of changes in interatomic couplings during stretching. The integrated crystal orbital Hamilton population (ICOHP) values evaluate the bonding strength between the corresponding atoms. Figures 4d-4e shows that the broken symmetry in CALF-20 leads to independent changes in the chemical bonds formed between Zn atoms and its coordinated atoms. Figure 4e evidences that the interactions between Zn and the oxalate ligands change only slightly in the whole strain range. Notably, the structural differences before/after the strain-softening point are mainly due to the different orientation of the oxalate ligands, as shown in Fig. 4d. The negative ICOHP analysis revealed that the interactions between the individual O or N atoms of the ligand and the Zn metal are significantly altered before and after the phase transition, while the interactions between the overall Zn and

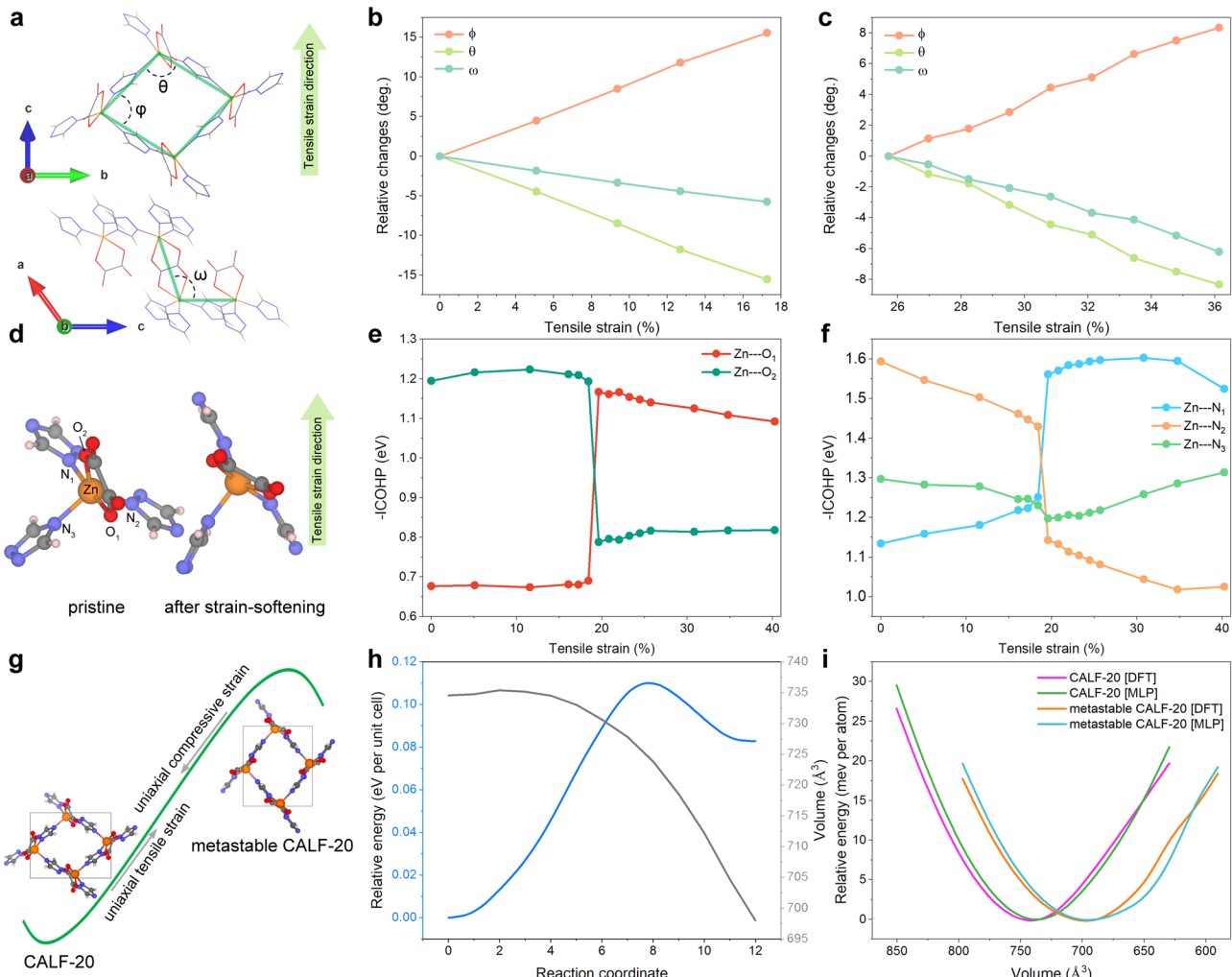

**Fig. 4 | Strain-induced phase transition of CALF-20. a** Scheme of CALF-20 along the **bc** (top) and **ac** (bottom) planes, where the angle $\phi$ and $\theta$ denote the angle between adjacent triazolate ligands in the lozenge lattice of CALF-20, and the angle $\omega$ represents the angle between the triazolate and oxalate ligands. Relative angle changes ($\phi$, $\theta$, and $\omega$) for the applied tension strain along the [001] direction (**b**) before strain-softening and (**c**) after strain-softening, respectively. **d** Typical CALF-20 structure component containing three 1,2,4-triazole and one oxalate functions. The five atoms coordinated to Zn metal are labelled (left: pristine CALF-20; right: CALF-20 after strain-softening. Colour code: Zn, orange; N, blue; O, red; C, grey; H,

white. **e, f** Negative integrated crystal orbital Hamilton population ($-$ICOHP) values reported as a function of the strain applied along [001] direction for various Zn-coordinated atom pairs. **g** Schematic illustration of the CALF-20 phase transition implying uniaxial tensile strain. **h** Energy landscape of the phase transition between the pristine structure and a metastable phase identified by generalised solid-state nudged elastic band (SS-NEB) approach[73]. Change of unit cell volume (Å³) relative to is given by black lines. **i** Unit-cell volume *vs* energy plot for pristine structure and metastable phase from DFT and MLP-MD simulations, respectively.

the two ligands vary only slightly, as shown in Fig. 4e, f. This is because during the stretching process along [001] direction, the main deformation is the bending of the Zinc triazolate grid, and the oxalate ligand acts as a hinge during the deformation process. Therefore, CALF-20 is demonstrated to exhibit a distinct two-step elastic deformation behaviour when stretched along the [001] direction, as well as a switching from auxetic to non-auxetic behaviour. This phenomenon is rarely reported in crystal[55], indicating that CALF-20 may be promising in the fields of engineering materials, motion memory devices and strain sensors among others[56,57].

The strained crystal structure is inherently unstable at ambient pressure. We therefore considered the structure after the strain-softening point and re-optimised it at the DFT level. Interestingly, the total energy of such configuration at 0 K was found to be only 83 meV unit-cell[-1] higher than that of the pristine structure. Therefore, this phase (termed metastable phase) is rearranged by applying uniaxial strain, showing strain-induced reversible phase transition behaviour, as shown in Fig. 4g. These calculations indicate that the energy barrier for the phase transition from pristine to metastable CALF-20 is only 0.11 eV unit-cell[-1], the energy barrier for the reverse phase transition being even lower (0.03 eV unit-cell[-1]) (Fig. 4h and Supplementary Movie 4). This suggests that the identified structure is indeed a metastable phase that can readily switch back to the pristine structure by overcoming a very low energy barrier. Interestingly, this prediction is in line with the conclusions drawn by a very recent experimental work published along the finalisation of this paper with the evidence of the phase (labelled β-CALF-20) formed when the material is exposed to humidity[58]. Notably, the structural parameters of this experimentally determined β-CALF-20 phase is in excellent agreement with that found for our predicted metastable phase as shown in Fig. 4g and Supplementary Table 4, paving the way towards an indirect validation of our computational findings. Furthermore, our calculations show that, the developed MLP can not only adequately predict both the mechanical properties and the deformation mechanism of the pristine CALF-20 within first-principles precision, but also predict the energy landscape of metastable phase accurately, as shown in Fig. 4i. Particularly considering the fact that such metastable phase has not been incorporated in the initial data set and MLP training, this suggests that the developed MLP can serve as a general-purpose potential for CALF-20.

To summarise, high-accuracy DFT and MLP-MD simulations enabled to systematically explore the flexible behaviour of CALF-20. This MOF was demonstrated to exhibit a coexistence of multiple negative phenomena simultaneously, including NAC, NPR and NTE, resulting in a unique dynamic response of the structure upon temperature and pressure stimuli. From a methodology standpoint, the development of an accurate MLP for the MOF framework was proved to be a key strategy to predict the mechanical properties at finite temperature with comparable numerical accuracy to DFT calculations and a much lower computational cost. Decisively CALF-20 was shown to possess a very unique strain-softening behaviour along the [001] direction with two nearly linear strain-stress curves under the ideal tensile strain applied, demonstrating an uncommon elastic deformation during the whole extension process. Importantly, the fracture strain of CALF-20 was determined to be 27% along the [001] direction at room temperature, a magnitude nearly twofold greater than the hitherto reported value for a MOF, specifically 14% in the case of Ni-TCPP[48]. This highlights that this material is very promising for biomechanical systems, flexible electronics and nanomechanical devices.

## Methods

### DFT calculations
All DFT calculations were carried out using the Vienna Ab-initio Simulation Package (VASP) code (Version: 5.4.4)[28]. The projector augmented wave (PAW) potential and the Perdew-Burke-Ernzerhof (PBE) exchange-correlation functional was adopted[59,60]. An energy cutoff of 650 eV and a Monkhorst-Pack $4 \times 3 \times 4$ $k$-point grid[61] was chosen to ensure convergence of total energy and forces below $10^{-5}$ eV and $10^{-3}$ eV/Å, respectively. We also used the DFT-D3 Van der Waals (vdW) correction[62]. The computed lattice parameters of the pristine CALF-20 were found in good agreement with the experimental values (Supplementary Table 5). The impact of $CO_2$ adsorption on the CALF-20 structure and its mechanical properties was also evaluated. More details can be found in Supplementary Note 4 and Supplementary Figs. 36–39.

### Dataset preparation
In order to obtain the training data for the MLPs introduced in this study, finite-temperature AIMD simulations were performed using the VASP code. These simulations involved > 30,000 snapshots within a $2 \times 2 \times 2$ supercell of CALF-20, using a time step of 0.5 fs. Brillouin zone sampling was performed using a Monkhorst-Pack $k$-point grid[61] of size $1 \times 1 \times 1$. All AIMD simulations adhered to the NVT ensemble framework, incorporating the Nosé-Hoover thermostat[63] to maintain a constant temperature of 100 ∼ 800 K.

### MLP development
To construct the MLP for CALF-20, the DeepMD-kit code (version 2.0.1)[64] was employed with the DeepPot-SE model[64,65]. The size of the embedding network was set to {25, 50, 100}, while the fitting section consisted of {240, 240, 240}. Both networks utilise the ResNet architecture. ResNet architecture is utilised in both networks[66]. During the training process, a cutoff distance of 7.9 Å was applied, and a smoothing value of 2.1 Å was used. To ensure model robustness, the training and test data were allocated in a 3:1 ratio, effectively mitigating the risk of overfitting.

### MLP-MD simulations
The MLP derived from the DFT was implemented in the following MLP-MD simulations through integration with the DeepMD-kit interface coupled with the LAMMPS code[64,67]. In the MLP-MD simulations, the trained model served as a pair style within the LAMMPS framework, enabling the computation of both energy and force profiles during the MD simulations. The phonon dispersion spectra were calculated under strict convergence criteria with the Phonopy code, by using LAMMPS as the calculator[35,67].

### Classical MD simulations
MD simulations employing classical force fields were executed using the LAMMPS code[67]. CALF-20 was treated as fully flexible with the potential parameters and 12-6 Lennard-Jones (LJ) site contributions from the universal force field (UFF)[68]. The computation of 12-6 LJ parameters involved the application of Lorentz-Berthelot mixing rules[69]. The distance cutoff for LJ interactions was set as 12 Å. Electrostatic interactions were determined using the Ewald summation method[70], with a tolerance level of $10^{-6}$. Atomic charges were determined utilising the DDEC6 method[71]. To accommodate the simulation, a supercell with dimensions of $3 \times 3 \times 3$ was employed, thereby ensuring that the box size was twice the cutoff radius.

More detailed methodologies are provided in the Supplementary Information.

## Data availability
The data used in this study are available in the Zenodo database in ref. 72. Source data are also provided in this paper. Source data are provided with this paper.

## Code availability
The primary packages utilized in this article including VASP[28], DeepMD-kit (https://github.com/deepmodeling/deepmd-kit)[64], and

LAMMPS[67]. Detailed information about the license and the user manual can be found in the abovementioned articles and on their websites.

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

## Acknowledgements

The computational work was performed using HPC resources from GENCI-CINES (Grant A0140907613).

## Author contributions

D.F. and G.M. designed the research. D.F. and S.N. carried out the simulations. D.F., S.N. and G.M. wrote the manuscript. G.M. supervised the research.

## Competing interests

The authors declare no competing interests.
