## [Peer Review File · Nature Communications]

REVIEWER COMMENTS

Reviewer #1 (Remarks to the Author):

While CALF-20 has become an industrial-scale CO₂ sorbent, a comprehensive understanding of its atomistic mechanical and thermal properties in operational conditions remains limited. Gaining insights into how CALF-20 responds to changes in pressure and temperature is not only of fundamental interest but also holds great significance for its practical use in a diverse range of operational conditions. In this study, Maurin and coauthors have developed a versatile machine-learned potential (MLP) designed specifically for the CALF-20 MOF framework. This MLP enables precise predictions of CALF-20's thermodynamic and mechanical properties at finite temperatures, with a degree of accuracy comparable to first-principles calculations. Therefore I would like to support this paper to be published. The comments listed as follows are some minor issues that can be addressed:

1. It would be beneficial to include a discussion of the structural characterization and flexibility of CALF-20 as observed in experimental studies. This background information would provide valuable context for the research conducted in this study.
2. Additionally, could the authors perform a structural analysis of CALF-20 at various temperatures? CALF-20 is anticipated to display negative thermal expansion (NTE) behavior at lower temperatures. It would be valuable to investigate how structural flexibility correlates with this characteristic at different temperature regimes.
3. CALF-20's response to tensile strain along the [001] direction is interesting. However, Figure 3g, illustrating the structural distinctions between auxetic and non-auxetic behaviors, lacks clarity. A revised version of the figure is needed for better comprehension.
4. The deformation of CALF-20 along the [001] direction was observed to trigger the formation of a newly metastable phase, akin to the recently reported β -CALF-20 phase. A comparison between these structures would be beneficial. Can authors use this method to discover more metastable structures of CALF-20?
5. Performing experimental tests to validate the predicted mechanical properties of CALF-20 through simulations would not only enhance the study but also serve as a means to validate the accuracy of the methodology employed. Authors might need to consider it.

Reviewer #2 (Remarks to the Author):

This manuscript describes calculations of the thermal, elastic, and mechanical properties of CALF-20 metal-organic framework using molecular dynamics based on machine learning potentials developed from DFT calculations.

CALF-20 is a promising MOF for CO₂ capture, with high selectivity of CO₂ over N₂ and over H₂O under flue gas conditions. It also has been reported to have negative thermal expansion and so-called negative area flexibility.

A major outcome is the finding that CALF-20 has negative area compression and thermal expansion up to 1 GPa. Two-step elastic deformation behavior was observed along [001] direction was ascribed to a phase transition.

I think is an overall strong study that could be suitable for publication, and I have a few minor comments on the technical aspects (see below). The one major critique is that the authors do not consider the effect of CO₂ in their simulations. Doing so would elevate the impact of their study. In other words, how would the thermal and mechanical properties change in the presence of CO₂?

Specific technical questions:

The authors state that this system could be attractive for mechanical and thermal sensing. Can the authors quantify how attractive this system might?

How do the authors validate that the ML potential is correctly predicting the behavior at finite temperatures? The data were generated starting with strained and unstrained configurations and in each case, an MD simulations in the NVT ensemble were performed, however the duration of the simulations were not specified, and it is not clear that these simulations are uncorrelated. The information on the length of their simulation during data generation and the specific temperature at which they performed their AIMD would be important for reproducibility.

In the isothermal compression studies (in Fig 1, comparing $T=0$ and $T=300\text{K}$), it looks like going from 0K to room T at 0GPa , the, the a decreases with T , b increases with T and c increases with T . Instead, in the thermal expansion studies, a increases with T , b decreases with T , while c is nearly constant. Could the authors comment on this?

The definition of compressibility should have $-1/i (di/dp)$.

In Fig. 1h, 1i, the b and c lattice parameters, as well as the angle β , computed by the MLP fluctuate noticeably off their established trends at 0.4GPa , before recovering and continuing on trend at higher pressures. Could the authors explain why?

Reviewer #3 (Remarks to the Author):

This manuscript presents as yet unstudied properties of a benchmark MOF material. The mechanical and thermal properties are unusual and can be used to derive added value for potentially a wide range of application and processes given the established stability and scalability of CALF-20.

The manuscript results are presented in detail and the authors present a logical and clear delineation of thought and experiment. I do feel the manuscript could be enhanced by adding more context on the specific values quoted for certain properties, especially for a general journal. For example, "a-axis (170TPa^{-1}) (see Fig. 1b), while both a and b axis show a negative contribution (maximum of -24.32TPa^{-1} and -32TPa^{-1} respectively).", "the most striking giant NAC phenomenon along a and b axis continues monotonically up to 0.3GPa . The NAC behavior is much more marked along the a -axis with -188.77TPa^{-1} strain tensor eigenvector compared to that along the b -axis (-75.52TPa^{-1})". Comparisons are made with other MOF materials but, again for a general, open access journal, I think there could be broader comparison outside the MOF family, perhaps to actual deployed compounds for real applications.

The manuscript has a "results" and "discussion" section but the discussion section is really conclusions. It would be better to rename it as such and add more discussion to a new "Results and Discussion" section. This can include more benchmarking beyond stating a name and a related value.

There are some typos, e.g. in the introduction, "making the whole testing process far to be trivial."

Reviewer #4 (Remarks to the Author):

In this paper, the authors developed a machine-learned potential (MLP) for the important metal-organic framework (MOF) CALF-20 and used it to investigate the mechanical and thermal behavior of this MOF. CALF-20 is an important material due to recent demonstration of its utility in CO_2 capture, and I think this work will be of interest to many researchers. In addition to reporting some interesting mechanical behavior of this material, the MLP itself will be useful to other researchers. I recommend the paper for publication after the authors address the following points.

1. It is not completely clear to me if the authors have made the MLP model available. I looked at the github page and it says, "To run molecular dynamics simulations efficiently using MLP, it is recommended to use compressed model file. Please contact the author if you need compressed

file, as the upload file size is limited to 25 Mb." So, it appears that the model is not really available, which will limit the impact of the work. I encourage the authors to put the full model somewhere online. Given the file size limitations of github, another site such as zenodo might be another option.

2. On page 8, the authors report that CALF-20 exhibits negative thermal expansion, but they also write that, "a nearly negligible unit cell volume change versus temperature is obtained." Negligible volume change versus temperature sounds contradictory to negative thermal expansion. This should be clarified.

3. I found the discussion of Figure 4d and 4e difficult to follow. What does the ICOHP tell us (physically)?

4. At the top of page 9, "Fig. 3c" should be "Fig. 2d."

5. Some of the labels in the figures are small and difficult to read.

RESPONSE TO REVIEWERS' COMMENTS

Reviewer #1

General Comment: *While CALF-20 has become an industrial-scale CO₂ sorbent, a comprehensive understanding of its atomistic mechanical and thermal properties in operational conditions remains limited. Gaining insights into how CALF-20 responds to changes in pressure and temperature is not only of fundamental interest but also holds great significance for its practical use in a diverse range of operational conditions. In this study, Maurin and coauthors have developed a versatile machine-learned potential (MLP) designed specifically for the CALF-20 MOF framework. This MLP enables precise predictions of CALF-20's thermodynamic and mechanical properties at finite temperatures, with a degree of accuracy comparable to first-principles calculations. Therefore I would like to support this paper to be published. The comments listed as follows are some minor issues that can be addressed:*

General Reply: We thank the reviewer for the very positive opinion of our paper and the recommendation to publish it in *Nat. Commun.*

Comment 1: *1. It would be beneficial to include a discussion of the structural characterization and flexibility of CALF-20 as observed in experimental studies. This background information would provide valuable context for the research conducted in this study.*

Reply 1: We thank the reviewer for the comment. It is indeed important to state that in contrast to other well-established MOFs, *e.g.* MIL-53, ZIFs, *etc.* that have been intensively explored in terms of mechanical properties and stimuli-induced structural changes, neither the mechanical nor the thermal behaviours of the more recent CALF-20 have never been explored so far from both experimental and computational standpoints. We also carefully checked the literature during the revision of the paper and we did not find any recent published works on this topic. Indeed this motivated us to explore the flexibility of this MOF first using advanced molecular simulations and we strongly believe that our publication reporting intriguing structural properties of

CALF-20 will further stimulate in-depth experimental investigations. As stated in the last section of the paper, the only experimental paper evoking the flexibility of CALF-20 we are aware of is an experimental work published during the evaluation of the paper on the formation of a new phase, labelled β -CALF-20, upon exposure to humidity. The discussion of this very recent experimental work is now expanded in the revised paper, paving the way towards an indirect validation of our simulation work evidencing a similar strain-induced metastable phase for the empty MOF.

To address this comment carefully, the following sentences were added on pages 2-3 of the main text:

“However, to date, the mechanical/thermal behaviours of CALF-20 have been overlooked both experimentally and theoretically. Although several studies have been reported on the competitive adsorption of CO₂ and H₂O in CALF-20 and derivatives¹⁹⁻²¹ the lack of understanding on the pressure/thermal-induced structure evolution of CALF-20 is critical in the context of using this MOF under diverse working conditions.”

The associated section was completed with the following sentences on page 15-16 of the main text along with new data on Figure 4g and Supplementary Table S4:

“Interestingly, this prediction is in line with the conclusions drawn by a very recent experimental work published along the finalization of this paper with the evidence of a new phase (labelled β -CALF-20) formed when the material is exposed to humidity.⁵⁹ Notably, the structural parameters of this experimentally determined β -CALF-20 phase is in excellent agreement with that found for our predicted new metastable phase as shown in Fig. 4g and Supplementary Table S4, paving the way towards an indirect validation of our computational findings. Furthermore, our calculations show that, the developed MLP can not only adequately predict both the mechanical properties and the deformation mechanism of the pristine CALF-20 within first-principles precision, but also predict the energy landscape of metastable phase accurately, as shown in Fig. 4i. Particularly considering the fact that such metastable phase has not been

incorporated in the initial data set and MLP training, this suggests that the developed MLP can serve as a general-purpose potential for CALF-20.”

Supplementary Table S4 | Structural parameters of our strain-induced metastable CALF-20 new phase and β -CALF-20 reported previously.³

	a	b	c	α	β	γ	V
	[Å]	[Å]	[Å]	[deg.]	[deg.]	[deg.]	[Å ³]
Exp.	9.279	7.934	10.039	90.000	109.891	90.000	694.972
PBE-D3	9.297	8.062	9.901	90.000	109.848	90.000	698.020

Comment 2: *Additionally, could the authors perform a structural analysis of CALF-20 at various temperatures? CALF-20 is anticipated to display negative thermal expansion (NTE) behavior at lower temperatures. It would be valuable to investigate how structural flexibility correlates with this characteristic at different temperature regimes.*

Reply 2: We thank the reviewer for raising this point. In the paper we verified the negative thermal expansion phenomenon using two different theoretical approaches:

- (i) we first used quasi-harmonic approximation (QHA) approach [Dove, M. T. **1993**. *Introduction to lattice dynamics; Scr. Mater.*, **2015** 108, 1-5] interfaced with the developed MLP to explore the thermal expansion properties of CALF-20 at different temperature regions (0-400K). We thus revealed the NTE behaviour, as shown in Fig. 2a of the main text. Owing to the high-quality of MLP, highly-accurate force constant matrix can be readily obtained, while traditional methods based on DFT calculations require high computational cost. It should also be noted that the QHA approach is a well-established theoretical method [*Phys. Rev. B*, **1999**, 60, 7234; *npj Comput.*

Mater., **2017**, 3(1), 44; *Nat. Commun.*, **2020**, 11, 4430 *etc.*, including predicting and explaining the thermal behaviors] and has been applied to the study of the thermal expansion properties of many different material systems including MOFs [S. Rogge, *Chapter 3 of Mechanical Behaviour of Metal–Organic Framework Materials*, 2023, 113-204; *PRX Energy*, **2023**, 2, 023005, *etc.*].

- (ii) Secondly, in order to calculate the structural changes of CALF-20 at different temperatures *more realistically*, we performed MLP-based MD simulations in the *NPT* ensemble (1 bar pressure) at room temperature, as shown in Figs. 2b-2c of the main text. It is clear that, as the temperature increases, the structural parameters of CALF-20 show anisotropic changes, with the *b*-axis dimension showing an abnormal decrease with increasing temperature, in line with the NTE property. It is also to be pointed out that the traditional *NPT* MD simulation procedures are based on the empirical force field method, which are highly sensitive to the selected force field parameters [*J. Phys. Chem. C*, **2008**, 112, 15, 5795–5802; *J. Mater. Chem. A*, **2019**, 7, 24019–24026]. In this work, we inventively used a high-accuracy MLP to perform a 1-ns long time MD *NPT* equilibrium of system, and finally obtained conclusion consistent with the QHA results.

Therefore, our advanced theoretical simulation methods support our conclusions very-well. We believe that such property can be detected in subsequent experiment. Especially considering the high potential of this MOF for industrial application, our work will surely attract the interest of experimental scientists to conduct research in this field.

Comment 3: *CALF-20's response to tensile strain along the [001] direction is interesting. However, Figure 3g, illustrating the structural distinctions between auxetic and non-auxetic behaviors, lacks clarity. A revised version of the figure is needed for better comprehension.*

Reply 3: We thank the referee for the comment. To address this comment, we redraw Figs. 3f-3g in the revised paper to clarify the structural distinctions between auxetic and non-auxetic behaviours (Fig. 3g) revealed by the plot of the volume changes of CALF-20 (Fig. 3f) when tensile strain is applied along the [001] direction.

The revised figure incorporated on page 11 of the manuscript is shown below:

Fig. 3f. | MLP-MD derived volume changes of CALF-20 at 298.15 K upon application of tensile strain along the three directions. The dotted line represents the unit-cell volume of the pristine CALF-20 structure. The two independent auxetic and non-auxetic phenomena upon stretching along the [001] direction are also highlighted.

Fig. 3g. | Schematic illustration of the CALF-20 structural evolution upon tensile strain applied along the [001] direction. The auxetic and non-auxetic behaviours correspond to the two regimes observed in Fig. 3f.

Comment 4: *The deformation of CALF-20 along the [001] direction was observed to trigger the formation of a newly metastable phase, akin to the recently reported β -CALF-20 phase. A comparison between these structures would be beneficial. Can authors use this method to discover more metastable structures of CALF-20?*

Reply 4: We appreciate this constructive comment. We fully agree that the strain-induced metastable phase our calculations reveal for the empty CALF-20 is akin to the very recent β -CALF-20 phase published upon exposure to humidity as now stated in the revised version of the manuscript following the Comment 1 of reviewer 1 (see above) along with the incorporation of Table S4 and Figure 4g. This very recent experimental work delivers an indirect validation of our computational methodology and prediction accuracy.

Regarding the reviewer's question about discovering more metastable phase structures of CALF-20 using this approach, it is not feasible since the tensile strain of more than 27% is already almost twice higher than that reported so far for a MOF (Ni-TCPP [*Adv. Mater.*, **2023**, 2210829]). Continuously increasing tensile strain will eventually induce a structure collapse. However, we trust that our proposed “*strain-induced phase transition*” approach can be systematically applied to the family of MOF materials to unravel other intriguing physical phenomena and structural behaviours.

Comment 5: *Performing experimental tests to validate the predicted mechanical properties of CALF-20 through simulations would not only enhance the study but also serve as a means to validate the accuracy of the methodology employed. Authors might need to consider it.*

Reply 5: We thank the referee for this comment. Indeed, as a theoretical group, we cannot unfortunately perform such experiments, but we strongly believe that our theoretical prediction will stimulate in-depth experimental characterization of this MOF by experts in the field. This holds even more true since part of our predictions have been indirectly validated experimentally by the recent work on the humidity-induced structural change of CALF-20 towards the new phase β -CALF-20.

Reviewer #2

General Comment: *This manuscript describes calculations of the thermal, elastic, and mechanical properties of CALF-20 metal-organic framework using molecular dynamics based on machine learning potentials developed from DFT calculations. CALF-20 is a promising MOF for CO₂ capture, with high selectivity of CO₂ over N₂ and over H₂O under flue gas conditions. It also has been reported to have negative thermal expansion and so-called negative area flexibility. A major outcome is the finding that CALF-20 has negative area compression and thermal expansion up to 1 GPa. Two-step elastic deformation behaviour was observed along [001] direction was ascribed to a phase transition. I think is an overall strong study that could be suitable for publication, and I have a few minor comments on the technical aspects (see below). The one major critique is that the authors do not consider the effect of CO₂ in their simulations. Doing so would elevate the impact of their study. In other words, how would the thermal and mechanical properties change in the presence of CO₂?*

General reply: We thank the reviewer for appreciating the important findings reported in our manuscript and for the very positive comments.

We agree that understanding the impact of CO₂ on the structural properties of CALF-20 is an important topic since this material is highly attractive for CO₂ capture as stated in the initial Science paper completed by many other papers [*Chem. Eng. J.* **2024**, 452, 139550; *Ind. Eng. Chem. Res.* **2024**, doi: 10.1021/acs.iecr.3c04266, *ACS Appl. Mater. Interfaces*, **2023**, 15, 48287-48295; *ACS Appl. Nano Mater. Interfaces*, **2023**, 6, 19963-19971]. Since the focus of the manuscript is on the intrinsic mechanical/thermal properties of CALF-20, in order to maintain the integrity of the manuscript, we could not include relevant discussion on the guest-induced structural changes of this MOF. However to carefully address the reviewer comment, we performed DFT calculations on the CO₂@CALF-20 system and provided our conclusions on the CO₂ dependence of the structure and mechanical properties of the MOF supported by relevant figures in the main text and Supplementary Material as detailed below.

Page 17 of the main text: “Impact of CO₂ adsorption on the CALF-20 structure and its mechanical properties was also evaluated. More details can be found in Supplementary Note4 and Supplementary Figs. S36-S39.”

“Supplementary Note5

Effect of CO₂ adsorption on the structure of CALF-20

DFT geometry optimization of the MOF structure loaded with different CO₂ concentrations up to the experimental saturation reported in the initial paper¹ (1~4 CO₂ molecules per unit cell) were carried at the same level of theory than used for the exploration of the MOF structure (PBE-D3) (Figs. S36-S38). We observed that when CO₂ loading increases, the lattice parameters of the CALF-20 show anisotropic evolution, arising from the anisotropic mechanical properties of the CALF-20 structure (see main text discussion). In particular, the *c*-parameter decreases the greatest when two CO₂ molecules per unit cell are loaded in the pores of CALF-20, and the unit cell volume decreases by 3.9% (cf. Figs. S37-S38). The primary origin of this contraction is that the confined CO₂ molecules at this loading occupy the central position of the pore of the CALF-20 structure, resulting in the internal chemical pressure generating the most pronounced volume shrinkage.

We next explored the effect of CO₂ for a given loading of 2 CO₂ per unit-cell on the stress-strain behaviour of the CALF-20 structure. As shown in Fig. S39, the introduction of CO₂ significantly improves the ductility of the CALF-20 structure, particularly along the [001] direction. The failure strain point increased from 18.4% (strain-softening point) to 26.9%, but simultaneously, we revealed that the presence of CO₂ prevents the formation of the strain-induced metastable phase evidenced for the empty case. Interestingly we reveal that the introduction of CO₂ also increases the auxetic region of the CALF-20 structure, and the structure retains its auxetic regime under a tensile strain of 23%.”

Supplementary Fig. S36 | Structural evolution of the CALF-20 unit-cell simulated by DFT calculations (using PBE-D3 functional) at different CO₂ loading (**a**, one CO₂ per unit-cell; **b**, two CO₂ per unit-cell; **c**, three CO₂ per unit-cell; **d**, four CO₂ per unit-cell).

Supplementary Fig. S37 | DFT-simulated relative unit-cell parameters change of CALF-20 loaded with different CO₂ concentration.

Supplementary Fig. S38 | DFT-optimized unit cell parameters (β angle and unit-cell volume) changes of CALF-20 loaded with different CO₂ concentration.

Supplementary Fig. S39 | (a) Tensile strain-stress response and (b) unit-cell volume

changes of the pristine CALF-20 during the application of tensile strain along the three directions from DFT calculations at 0 K. (c) Tensile strain-stress response and (d) unit-cell volume changes of the CALF-20 loaded with two CO₂ molecules per unit cell during the application of tensile strain along the three directions from DFT calculations at 0K.

Comment 1: *Specific technical questions: The authors state that this system could be attractive for mechanical and thermal sensing. Can the authors quantify how attractive this system might?*

Reply 1: Following this valuable suggestion, we made the following revisions in the manuscript as follows:

Page 9 of the main text: *“While this coefficient is comparable to that exhibited by MOF DUT-60 (-65.0 MK^{-1})⁴¹ it is twice higher than the values reported for other typical NTE MOFs, e.g. Ag(mim) (-24.5 MK^{-1}),⁴² and DUT-49 (-32.778 MK^{-1})⁴³. This suggests CALF-20 as a promising candidate for applications in the fields of ultrasensitive thermal sensing.”*

Page 13 of the main text: *“This suggests that the CALF-20 structure presents a high flexibility in conjunction with a good mechanical strength. In particular, the linear relationship between the angular changes and the applied tensile strain is still preserved above 30% strain as shown in Fig. 4c and Supplementary Movie S1. This MOF offers alternative solution to Mxene and graphene^{52,53} for ultra-sensitive strain-sensing applications since its predicted elongation range is lower than 5%.”*

Comment 2: *How do the authors validate that the ML potential is correctly predicting the behaviour at finite temperatures? The data were generated starting with strained and unstrained configurations and in each case, an MD simulations in the NVT ensemble were performed, however the duration of the simulations were not specified, and it is not clear that these simulations are uncorrelated. The information on the length*

of their simulation during data generation and the specific temperature at which they performed their AIMD would be important for reproducibility.

Reply 2: We agree with the reviewer that the validation of the MLP is highly important. In our study, we used the following criteria to cross-validate our MLP:

- 1) **Root mean square error (RMSE) of the energy and force compared to DFT calculations.** The MLP model has to be validated first by its accuracy in describing the energy and force initially obtained at the DFT level. This first validation stage is critical for reproducing the structural properties of the system of interest. [*Nat Commun.* 2020, 11, 5461] This strategy has been widely adopted in many different structural systems, including but not limited to bulk, two-dimensional, molecular systems, liquids, solid-liquid interfaces, *etc.* [*Phys. Rev. X*, **2018**, 8, 041048. *Phys. Rev. Lett.* **2021**, 126, 236001; *Proc. Natl. Acad. Sci. U.S.A.* **2021**, 118, e2110077118] According to our calculations, as shown in Fig. S5, the RMSE of the energy and force for CALF-20 are 0.068 meV atom⁻¹ and 0.023 eV Å⁻¹, respectively, indicating the very high accuracy of the developed MLP in describing the features of the CALF-20 system. More details on this discussion are included in Supplementary Note1.

Supplementary Fig. S5 | Machine learning potential (MLP) derived (a) energy

and (b) force vs the corresponding DFT values in the potential testing.

- 2) **Comparison of energy profiles calculated by DFT and MLP.** Theoretically, for a well-trained MLP, the calculated total energy change should be in good agreement with the results of the DFT calculations, thereby proving that the trained potential can well describe the many-body interactions between atoms. [*Nat. Commun.* **2022**, 13, 3733; *Nat. Commun.* **2023**, 14, 1008]. In particular, the capability to capture the lowest-energy/saddle point is crucial for subsequent MLP-based molecular dynamics simulations. As shown in Figs. S6-S7, our MLP for CALF-20 can successfully reproduce the energy profile of the equilibrium state as well as the energy profile under the application of mechanical pressure, demonstrating that the MLP can well describe the intramolecular interactions in the CALF-20 framework. More detailed discussion can be found in Supplementary Note1.

Supplementary Fig. S6 | Volume-energy curves for CALF-20 simulated by DFT and MLP calculations.

Supplementary Fig. S7 | DFT and MLP calculated total energy of the unit cell of CALF-20 under different applied mechanical pressures.

- 3) **Comparison of the radial distribution functions calculated by AIMD and MLP simulations.** This validation is crucial to ensure that the trained MLP can successfully describe the structural properties of the structure at finite temperature. This answers the reviewer's question: “*How do the authors validate that the ML potential is correctly predicting the behaviour at finite temperatures?*” Generally, for a given crystal structure, the interactions between the constitutive atoms can be statistically characterised by the radial distribution functions (RDFs) calculated for different atom pairs. If the RDFs computed based on MLP can match the results based on AIMD outputs, it means that the trained MLP can accurately describe the dynamic properties of structures at finite temperature. Such comparisons have been successfully applied to crystalline structures of different dimensions, as well as to small molecules and liquids etc. [*Phys. Rev. Lett.* **2018**, 120, 143001; *Nature*, **2020**, 585, 217-220; *Nat. Commun.* **2021**, 12, 766]. As shown in Fig. S13, the calculated RDFs using our trained MLP are in line with that obtained by AIMD

calculations, which highlight that our MLP can describe well the structural properties of CALF-20 at finite temperature. More detailed discussion can be found in Supplementary Note1.

Supplementary Fig. S13 | (a, b) Statistical Radial Distribution Functions (RDFs) calculated for the atom pairs of CALF-20 at 300 K by using MLP and AIMD simulations, respectively.

We also agree with the reviewer that it is important to specify the length of the AIMD simulations in the paper. According to our testing (as shown in the figure below, Fig. R1), for the pristine CALF-20 structure, even at a the very high temperature of 800K, the structure can reach equilibrium within ~ 1 ps, therefore our total AIMD simulation run was considered for about 2 ps.

Fig. R1 | Evolution of temperature and total energy of the CALF-20 structure ($2 \times 2 \times 2$ supercell) with AIMD simulation time. The simulation employs NVT ensemble and the simulation temperature is set to 800 K.

We added relevant content to the Supplementary Note3 of the Supplementary Material as shown below (Page 55 of the Supplementary Material):

“It should be noted that in order to improve the structural sampling, AIMD simulations of the CALF-20 supercell structure were performed using different conditions as summarized in Table S6. The simulation time for each AIMD simulation was around about 2 ps.”

To facilitate the reproducibility of our findings, we included all the AIMD computational results and uploaded it to the zenodo platform (the link can be found below), as well as the necessary input files. We believe that all the data we share and the detailed computational methodology description in the Supplementary Material of the paper will be very useful for researchers who are interested to use our MLP and expand this work towards other avenues. We also believe that the data we share will help interested scientists to apply our proposed computational schemes to the materials

they are curious to study, thereby promoting the application of this multiscale simulation computational method to different materials.

GitHub: https://github.com/agrh/Data_Repository_CALF-20_MLP

Zenodo: <https://doi.org/10.5281/zenodo.10650655>

The Data availability section of the main text has been modified accordingly

“All data needed to evaluate the conclusions in the paper are present in the paper and/or the Supplementary Information. The data related to this article can be accessed online at https://github.com/agrh/Data_Repository_CALF-20_MLP and <https://doi.org/10.5281/zenodo.10650655>. Source data are also provided in this paper.”

Comment 3: *In the isothermal compression studies (in Fig 1, comparing $T=0$ and $T=300K$), it looks like going from 0K to room T at 0GPa, a decreases with T , b increases with T and c increases with T . Instead, in the thermal expansion studies, a increases with T , b decreases with T , while c is nearly constant. Could the authors comment on this?*

Reply 3: We thank the reviewer for the comment. To clarify Fig. 1d reports the structural parameters obtained by the DFT geometry optimizations at 0K performed with *no applied mechanical pressure* (0.0 GPa). In Fig. 1h, the structural parameters are averaged over the trajectories collected along the MLP-MD run at 298,15K (see figure below, Fig. R2). Therefore, these two sets of lattice parameters cannot be directly compared. These data reported in Fig.1h should be compared with the data shown in Fig. 2b also obtained using MLP-MD simulations at different temperatures.

Fig. R2 | (a, b) Evolution of the CALF-20 structural parameters with simulation time based on MLP-MD simulations at 298.15 K (*NPT* ensemble, 1 bar pressure).

Comment 4: *The definition of compressibility should have $-1/i$ (di/dp).*

Reply 4: We thank the referee for the reminder. The corresponding text in the manuscript was modified accordingly (Page 6 of the main text):

“Compressibility of material is typically denoted as the relative rate of dimension changes with pressure at a fixed temperature, $\kappa = -\frac{1}{i} \frac{di}{dp}$, where i can be assigned as volume, area, or linear compressibility, respectively.⁹”

Comment 5: *In Fig. 1h, 1i, the b and c lattice parameters, as well as the angle beta, computed by the MLP fluctuate noticeably off their established trends at 0.4 GPa, before recovering and continuing on trend at higher pressures. Could the authors explain why?*

Reply 5: We thank the reviewer for raising this point. This is mainly because our total MLP-MD simulation time is 1 ns for each pressure, and the statistics on the structural parameters are performed over the last 500 ps. For the MLP-MD simulation at 0.4 GPa, as shown in the figure below, there is a sudden change in the structure around 800 ps

(Figure R3) and this is the reason why the statistical lattice curves show a large fluctuation region. But for other conditions such as 1.0 bar and 1.0 GPa pressure, there is no obvious structural parameter changes during the simulation (see Fig. R4).

Fig. R3 | (a, b) Evolution of the CALF-20 structural parameters with simulation time based on MLP-MD simulation at 298.15 K under 0.4 GPa pressure.

Fig. R4 | (a, b) Evolution of CALF-20 structural parameters with simulation time based on MLP-MD simulation at 298.15 K under 1.0 GPa pressure.

However, this is reasonable because the MLP-MD based *NPT* simulation is a statistical result, and the *bc* plane corresponds to the more flexible rhombic-shaped zinc triazolate grid while the *a*-axis is the pillared oxalate ligands direction (which is relatively rigid), as shown in Fig. 1a of the main text. Therefore, the fluctuations for the *a*-axis are relatively small during the MLP-MD simulations, while the statistical errors for the *b*-axis and *c*-axis are larger, especially when the phase transition occurs (0.3 ~ 0.4 GPa for example). To account for this, we modified the relevant part of the statement (Page 7 of the main text):

“Figs. 1h-1i report the MLP-MD derived evolution of the CALF-20 lattice parameters at room temperature under different pressures. These simulations evidenced that both a- and b-axis dimensions decrease with increasing pressure below 0.30 GPa in line with the NAC phenomenon revealed by DFT calculations at 0K. An apparent phase transition is equally observed around 0.30-0.4 GPa (Figs. 1d and 1h). The fluctuations of the b- and c-axis dimensions are found to be larger than that of the a-axis one within the phase transition region as shown in Fig. 1h. This is mainly due to the intrinsic anisotropic flexibility of the CALF-20: its bc-plane corresponds to the more flexible zinc triazolate lattice, while the a-axis is aligned with the more rigid pillared oxalate ligands. Moreover, when pressure exceeds 0.50 GPa, a- and b-axis dimensions remain almost unchanged upon pressure increase, while the c-axis dimension decreases linearly (Fig. 1h).”

Reviewer #3

General Comment: *This manuscript presents as yet unstudied properties of a benchmark MOF material. The mechanical and thermal properties are unusual and can be used to derive added value for potentially a wide range of application and processes given the established stability and scalability of CALF-20.*

The manuscript results are presented in detail and the authors present a logical and clear delineation of thought and experiment. I do feel the manuscript could be enhanced by adding more context on the specific values quoted for certain properties, especially for a general journal.

General Reply: We are grateful to the reviewer for the positive opinion and for posing several valuable suggestions to improve the quality of our manuscript.

Comment 1: *For example, “a-axis (170 TPa⁻¹) (see Fig. 1b), while both a and b axis show a negative contribution (maximum of -24 32 TPa⁻¹ and -32 TPa⁻¹ respectively).”, “the most striking giant NAC phenomenon along a and b axis continues monotonically up to 0.3 GPa. The NAC behaviour is much more marked along the a-axis with -188.77 TPa⁻¹ strain tensor eigenvector compared to that along the b-axis (-75.52 TPa⁻¹)”. Comparisons are made with other MOF materials but, again for a general, open access journal, I think there could be broader comparison outside the MOF family, perhaps to actual deployed compounds for real applications.*

Reply 1: We agree with the reviewer that for the general journal like *Nat. Commun.*, we need to consider a comparison with a broader range of materials. We modified the following sentence on page 6 of the manuscript:

“Remarkably, in contrast to other conventional uniaxial NLC materials, CALF-20 exhibits a biaxial (or in-plane) NLC behavior, and its compressibility is higher than that of the most extreme existing NLC inorganic materials including typical ferroelastics (-0.137~-4.3 TPa⁻¹),³¹ Ag₃[Co(CN)₆]-I (K_c = -76 TPa⁻¹)²⁹ and

[Ag(en)]NO₃-I ($K_c = -28.4 \text{ TPa}^{-1}$),³² as well as the emblematic flexible MIL-53 MOF material ($K_b = -27 \text{ TPa}^{-1}$).³³”

Comment 2: *The manuscript has a “results” and “discussion” section but the discussion section is really conclusions. It would be better to rename it as such and add more discussion to a new “Results and Discussion” section. This can include more benchmarking beyond stating a name and a related value.*

Reply 2: We thank the reviewer for the comment and indeed we agree that results and discussion sections should be added. In the revised version, we changed the corresponding headline and replaced the “Discussion” with “Conclusion”. In addition, we modified/added a discussion of the β -CALF-20 structure in section “Reversible strain-induced phase transition mechanism”, as follows (Pages 15-16 of the main text):

“Interestingly, this prediction is in line with the conclusions drawn by a very recent experimental work published along the finalization of this paper with the evidence of a new phase (labelled β -CALF-20) formed when the material is exposed to humidity.⁵⁹ Notably, the structural parameters of this experimentally determined β -CALF-20 phase is in excellent agreement with that found for our predicted new metastable phase as shown in Fig. 4g and Supplementary Table S4, paving the way towards an indirect validation of our computational findings. Furthermore, our calculations show that, the developed MLP can not only adequately predict both the mechanical properties and the deformation mechanism of the pristine CALF-20 within first-principles precision, but also predict the energy landscape of metastable phase accurately, as shown in Fig. 4i. Particularly considering the fact that such metastable phase has not been incorporated in the initial data set and MLP training, this suggests that the developed MLP can serve as a general-purpose potential for CALF-20.”

Comment 3: *There are some typos, e.g. in the introduction, “ making the whole testing process far to be trivial.”.*

Reply 3: We thank the reviewer for pointing out this typographical error. We modified the relevant sentence in the main text accordingly (Page 3 of the main text):

“The determination of MOF mechanical/thermal properties generally requires the deployment of sophisticated experimental techniques as well as the consideration of high-quality samples making the whole testing process far from trivial.”

We also modified other typographical errors by a careful proof-reading of the manuscript.

Reviewer #4

General Comment: *In this paper, the authors developed a machine-learned potential (MLP) for the important metal-organic framework (MOF) CALF-20 and used it to investigate the mechanical and thermal behaviour of this MOF. CALF-20 is an important material due to recent demonstration of its utility in CO₂ capture, and I think this work will be of interest to many researchers. In addition to reporting some interesting mechanical behaviour of this material, the MLP itself will be useful to other researchers. I recommend the paper for publication after the authors address the following points.*

General Reply: We greatly appreciate the positive comments of the reviewer. And we agree that the publication of this work will not only attract the interest of researchers in the field of MOFs, but also encourage theoretical scientists to consider the advantages of MPL for a broader range of porous materials. With the multiscale approach presented in the manuscript, one can systematically explore the mechanical/thermodynamic properties of the structure under different specific temperatures at the atomic level, which is not feasible with classical molecular dynamics simulations (low-accuracy) and at the first-principles level (computationally expensive).

Comment 1: *It is not completely clear to me if the authors have made the MLP model available. I looked at the github page and it says, "To run molecular dynamics simulations efficiently using MLP, it is recommended to use compressed model file. Please contact the author if you need compressed file, as the upload file size is limited to 25 Mb." So, it appears that the model is not really available, which will limit the impact of the work. I encourage the authors to put the full model somewhere online. Given the file size limitations of github, another site such as zenodo might be another option.*

Reply 1: We thank the reviewer for this suggestion. We uploaded all the original files to the zenodo platform to complement github. Changes were made to the relevant links:

GitHub: https://github.com/agrh/Data_Repository_CALF-20_MLP

Zenodo: <https://doi.org/10.5281/zenodo.10650655>

The Data availability section of the main text has been modified accordingly

“All data needed to evaluate the conclusions in the paper are present in the paper and/or the Supplementary Information. The data related to this article can be accessed online at https://github.com/agrh/Data_Repository_CALF-20_MLP and <https://doi.org/10.5281/zenodo.10650655>. Source data are also provided in this paper.”

All the source data are included in the above links, and we provide a very detailed explanation of the method in the supplementary material (see Supplementary Note3), which includes a variety of computational content, including DFT calculations, machine learning potential training, and empirical molecular dynamics simulations *etc.* We believe that the datasets and computational input files we have shared will make stimulate more works by the material science community.

Comment 2: *On page 8, the authors report that CALF-20 exhibits negative thermal expansion, but they also write that, “a nearly negligible unit cell volume change versus temperature is obtained.” Negligible volume change versus temperature sounds contradictory to negative thermal expansion. This should be clarified.*

Reply 2: We kindly thank the reviewer for this comment. Our conclusion is still valid and it does not contradict the negative thermal expansion property of CALF-20. Typically, for anisotropic materials such as CALF-20, negative thermal expansion can be referred as linear/area negative thermal expansion [see for instance *CrystEngComm*, **2014**, 16, 3498-3506] and the volume may increase or be almost unchanged as typically reported in this paper [*J. Am. Chem. Soc.* **2013**, 135, 17, 6411–6414]. Therefore, following the conclusions of our calculations, CALF-20 structure can be considered with almost zero volumetric thermal expansion, [*Nature*, **2003**, 425, 702-705; *Inorg. Chem.*, **2022**, 61, 18458–18465] but at the same time it has area negative thermal

expansion. However, we believe that the use of many different terms in the manuscript would introduce confusions to the readers and this explains why we have referred to it as negative thermal expansion in the main text. To address the reviewer comment, we modified the relevant statements in Page 9 of the main text:

*“We evidenced that the **b**-axis dimension decreases linearly with increasing temperature while the overall unit-cell volume remains almost constant, showing a unique NTE feature,⁴⁰ in excellent agreement with the prediction based on the QHA methodology.”*

Comment 3: *I found the discussion of Figure 4d and 4e difficult to follow. What does the ICOHP tell us (physically)?*

Reply 3: We thank the referee for this comment. The relevant discussion in this “Reversible strain-induced phase transition mechanism” section is to better understand the mechanism and origin of phase transitions exhibited by CALF-20 upon tension strain applied along the [001] direction. Figs. 4a-4c show the structural evolution of the CALF-20 framework with the applied tension strain along the [001] direction. The linear relationship of the framework changes can be clearly identified from these plots. While the -ICOHP analysis allows the investigation of changes in the interatomic bonding between atoms. For other crystal structures, their unconventional strain characteristics may be due to changes in the electronic structure (*i.e.* changes in bonding condition), which manifest as sudden changes in -ICOHP [*Nature*, **2013**, 496, 339–342; *Phys. Rev. B*, **2019**, 100.6, 060102]. However, for CALF-20, we found that its unconventional structural behaviour is due to the flexibility of the structure itself.

As shown in the figure below, before and after the phase transition, the -ICOHP value fluctuates within a certain range (highlighted by light-green) without significant sudden drop or increase. Therefore, from the analysis of the -ICOHP changes, the phase transition can be derived from the reciprocal shift in bonding roles between the responsible atoms (between Zn and O_{oxalate}, Zn and N_{triazolate} atoms, respectively) before

and after the strain-softening state. For example, in Figs. 3d-3e, before the strain-softening point, the -ICOHP value of Zn-O₁ is greater than that of Zn-O₂, but the reverse is observed after the strain softening point, with both remaining within a specified range without any sudden drop/increase. This shows that there are no obvious changes in chemical bonds after the phase transition meaning that the flexibility of the structure itself plays the predominant role.

Fig. R5 | Negative integrated crystal orbital Hamilton population (-ICOHP) values reported as a function of the strain applied along [001] direction for various Zn-coordinated atom pairs.

In order to clarify the discussion of the -COHP parts we revised the relevant section in the main text (Pages 13-14 of the main text):

“The integrated crystal orbital Hamilton population (ICOHP) values evaluate the bonding strength between the corresponding atoms. Figs. 4d-4e shows that the broken symmetry in CALF-20 leads to independent changes in the chemical bonds formed between Zn atoms and its coordinated atoms. Fig. 4e evidences that the interactions between Zn and the oxalate ligands change only slightly in the whole strain range. Notably, the structural differences before/after the strain-softening point are mainly due to the different orientation of the oxalate ligands, as shown in Fig. 4d. The negative ICOHP analysis revealed that the interactions between the individual O or N atoms of

the ligand and the Zn metal are significantly altered before and after the phase transition, while the interactions between the overall Zn and the two ligands vary only slightly, as shown in Figs. 4e-4f.”

Comment 4: *At the top of page 9, “Fig. 3c” should be “Fig. 2d.”*

Reply 4: We apologize for the typo, it was corrected accordingly on Page 9 of the main text).

Comment 5: *Some of the labels in the figures are small and difficult to read.*

Reply 5: We checked all the figures carefully, as they were all prepared as double-column figures arranged tightly, especially Fig. 4. Indeed, the labelling of the figure was not clear enough. In the new version, we have redrawn Fig. 3 and Fig. 4 as follows:

Fig. 3 | Unconventional strain-stress behaviour determined by DFT and finite temperature *NPT* MLP-MD simulations.

Fig. 4 | Strain-induced phase transition of CALF-20.

REVIEWERS' COMMENTS

Reviewer #1 (Remarks to the Author):

The authors have substantially revised the manuscript and addressed my previous concern fully. Therefore, I am delighted to recommend it for publication in Nature Communications at it is.

Reviewer #2 (Remarks to the Author):

I appreciate the authors' thorough response to my questions, and I can now recommend publication in Nature Communications.

Reviewer #3 (Remarks to the Author):

The authors have addressed my concerns. I am happy to recommend acceptance

Reviewer #4 (Remarks to the Author):

The authors have addressed my concerns about the paper, and I find their responses to the comments from the other reviewers to be satisfactory. I now recommend the paper for publication.

RESPONSE TO REVIEWERS' COMMENTS

Reviewer #1: *The authors have substantially revised the manuscript and addressed my previous concern fully. Therefore, I am delighted to recommend it for publication in Nature Communications as it is.*

Reviewer #2: *I appreciate the authors' thorough response to my questions, and I can now recommend publication in Nature Communications.*

Reviewer #3: *The authors have addressed my concerns. I am happy to recommend acceptance.*

Reviewer #4: *The authors have addressed my concerns about the paper, and I find their responses to the comments from the other reviewers to be satisfactory. I now recommend the paper for publication.*

Reply: We wish to thank all referees again for the time invested in the report and for the thoughtful comments.